# Microbial Transglutaminase Cross-Linking Enhances the Textural and Rheological Properties of the Surimi-like Gels Made from Alkali-Extracted Protein Isolate from Catfish Byproducts and the Role of Disulfide Bonds in Gelling

**DOI:** 10.3390/foods12102029

**Published:** 2023-05-17

**Authors:** Yan Zhang, Sam K.C. Chang

**Affiliations:** 1Coastal Research and Extension Center, Mississippi State University, Pascagoula, MS 39567, USA; 2Department of Food Science, Nutrition and Health Promotion, Mississippi State University, Starkville, MS 39762, USA

**Keywords:** rheological, TPA, disulfide bonds, electrophoresis, protein isolate, proteolysis, catfish byproducts

## Abstract

The texture of surimi-like gels made from the protein isolate extracted from catfish byproducts has been proven to be brittle and lack elasticity. To address this issue, varying levels of microbial transglutaminase (MTGase) from 0.1 to 0.6 units/g were applied. MTGase had little effect on the color profile of gels. When MTGase at 0.5 units/g was employed, hardness, cohesiveness, springiness, chewiness, resilience, fracturablity, and deformation were increased by 218, 55, 12, 451, 115, 446, and 71%, respectively. A further increase in added MTGase did not lead to any textural improvement. In comparison to the gels made from fillet mince, the gels made from protein isolate were still lower in cohesiveness. Due to the activated endogenous transglutaminase, a setting step enhanced the textural properties of gels made from fillet mince. However, because of the endogenous proteases-induced protein degradation, the setting step led to a texture deterioration of the gels made from protein isolate. Gels made from protein isolate showed 23–55% higher solubility in reducing solution than in non-reducing solution, suggesting the vital role of disulfide bonds in the gelation process. Due to the different protein composition and conformation, fillet mince and protein isolate exhibited distinct rheological properties. Sodium dodecyl sulfate-polyacrylamide gel electrophoresis (SDS-PAGE) showed the highly denatured protein isolate was susceptible to proteolysis and prone to disulfide formation during the gelation process. It also revealed that MTGase had an inhibitory effect on the proteolysis induced by endogenous enzymes. In view of the susceptibility of the protein isolate to proteolysis during gelation, future research should consider including other enzyme inhibitory agents in the presence of MTGase to improve the gel texture.

## 1. Introduction

Microbial transglutaminase (MTGase) has been widely used to improve the strength of gels made from various fish species. However, different myofibrillar proteins show a distinct polymerization degree and speed in response to MTGase [1]. It should be noted that the MTGase substrates, glutamine, and lysine residues in the muscle proteins vary remarkably among different fish species [2]. In addition, how protein structural changes affect the effectiveness of cross-linking reactions in terms of the optimal enzyme activity required has not been fully understood. It is well known that protein structural changes can be induced by various complex environmental factors to produce specific effects on protein functionality, and the extent of the effect depends on the intensity and time of exposure to these factors, including alkalinity/acidity, temperature (heating or freezing), pressure, and salt content. In addition, the status of the raw materials, storage, and processing, such as freshness and grinding, would also produce an impact. Thus far, our laboratory is the only laboratory that investigated the overall proteins isolated from finely ground catfish heads and bone frames; these materials have to be ground extensively into fine particles prior to extraction. Exposures to the high pH environment of purified catfish fillet myosin have been shown to cause a decrease in α-helix content [3]. However, this myosin study did not investigate the changes of other secondary structures as affected by pH shifts.

Our previous studies have proven that protein extraction from catfish byproducts with a high pH method increased mass yield, which was important for its commercial feasibility [4,5,6]. Our recent study [5] showed that compared to a mild-alkali (pH 8.5) extracted byproduct protein, higher-alkali (pH 11) extracted proteins caused a 21.5% decrease in α-helical structure. Furthermore, we have shown that protein isolated at pH 11 from catfish byproducts caused a 70% decrease in solubility when compared to a non-denatured protein extracted using a 0.6 M salt solution from byproducts [6]. The decrease in solubility was associated with the changes in protein molecular structures with an 18% decrease in α-helix, 47% increase in β-sheet, and 17% increase in random coil [6]. As a result, the gels made from the denatured protein isolate were firm but lacked elasticity [6].

Differently from the traditional pH-shift protein extraction from fish fillet mince used by some researchers [7,8], the catfish protein in the byproducts (heads and frames) from fillet processing was extracted with extensive grinding and long-time strenuous stirring, which resulted in a high degree of protein denaturation [6] since the raw byproduct protein had only approximately 44% helical structures as compared to native myosin helix content of about 80% [3,9]. In addition, the unfolded proteins after alkaline extraction may make it easily accessible to endogenous proteases. Our recently published data showed that after gelation of the protein isolate with a setting step, the TCA soluble peptides increased by 140% [10]. These specific characteristics of the highly denatured protein isolate should make it behave differently when MTGase is applied.

Our previous study proved that for the protein isolate extracted under the mild alkaline condition (pH 8.5), MTGase in the range of 0 to 4 units/g could improve the breaking force and deformation of the gels [5]. However, due to the higher alkalinity, the protein isolate yield extracted at pH 11 was 36% higher than that from pH 8.5, even though the protein structures were more affected at pH 11 as evidenced by a significant decrease in the α-helical structures from 36.6 to 28.8% [5]. Therefore, to better utilize the high pH isolated protein, there is a need to elucidate how MTGase at a lower range of reaction units (0–0.6 units/g) behaves in the more denatured protein isolated at pH 11. To date, no information is available with respect to the application of MTGase to the highly denatured catfish byproduct protein isolate. Therefore, the objectives of the current study are as follows: (1) Characterize the reactions of MTGase under a lower range of enzyme activity to improve the texture profile of gels made from highly denatured catfish protein isolate; (2) Explore the effects of disulfide bonds and MTGase on the gelation mechanism.

## 2. Materials and Methods

### 2.1. Materials and Preparation of Protein Isolate

Fresh catfish fillet and byproducts (frames and heads) were provided by a catfish processing plant (Harvest Select Catfish, Uniontown, AL, USA) and transported to the Experimental Seafood Processing Laboratory at Mississippi State University in an ice chest. The protein isolate was extracted at pH 11 and precipitated at pH 5.5 according to the recently reported procedures by our laboratory [10]. ACTIVA RM100 microbial transglutaminase (50 units/g) was obtained from Ajinomoto (Teaneck, NJ, USA).

### 2.2. Preparation of Gels

Gels were prepared according to Zhang and Chang with some modifications [6]. The pH and moisture of the protein isolate were adjusted to 7 and 80%, respectively. Then, 2% NaCl and a series of concentrations of MTGase (0.1, 0.2, 0.3, 0.4, 0.5, and 0.6 units/g) were added. The mixture was thoroughly comminuted to a homogenous paste with a pestle in a mortar. For comparison, gels were also prepared with catfish fillet mince. The catfish fillet was ground through a 4.5 mm plate with a LEM #32 grinder. With ice water and NaCl (2%) added, the ground mince was homogenized for 30 min into a paste using a Stephan mixer (Stephan Machinery GMBH, Hameln, Germany) at 4 °C under a vacuum. The final moisture content of the paste was also adjusted to 80%. The well-mixed pastes were stuffed into tubes (30 mm in diameter and 110 mm in length). For the optimal activity of MTGase, the protein isolate added with MTGase was heated using a setting method (40 °C for 30 min followed by 90 °C for 20 min). For comparison, protein isolate without MTGase and fillet mince were heated using both the setting method and direct heating method (90 °C for 30 min). During heating, a glass ball was placed on the top of each tube to avoid water evaporation. After heating, the tubes were immediately cooled in an ice water bath. All samples were stored at 4 °C in a walk-in cooler prior to analysis.

### 2.3. Color Analysis of Gels

The gel was cut into cylinders (30 mm in diameter and 15 mm in height) and each sample was measured twice in a glass sample cup. The gel made without MTGase added was used as a control. Before measurement, the sensor was standardized using a black glass and a white tile. CIE L*, a*, and b* values of the gels were measured in a ColorFlex EZ spectrophotometer (HunterLab, Reston, VA, USA). The below equation was used to calculate whiteness [11].
Whiteness = 100 − [(100 − L*)^2^ + b*^2^ + c*^2^]^1/2^(1)

### 2.4. Texture Profile Analysis

A TA.XT Plus texture analyzer was used to perform texture profile analysis. The method reported by Meng et al. [12] was adopted with slight modifications. The gels were equilibrated at room temperature and then cut into 15 mm cylindrical pieces. Two 75% compressions were performed using a plate with a diameter of 100 mm. The speed was set at 100 mm/min. Firmness, springiness, cohesiveness, resilience, chewiness, fracturability, and deformation were determined.

### 2.5. Dynamic Rheological Analysis

According to the method recently reported by our lab [6], temperature sweeps were conducted on a MCR 102 Anton Paar rheometer (Anton Paar GmbH, Graz, Styria, Austria) to monitor the sol-gel transition. To better depict the gelation process, the temperature was set according to the real temperature ramp during the setting and heating processes. The gap between the two 25 mm parallel plates was 1 mm. The shear frequency and shear strain were set at 0.5 Hz and 0.3%, respectively. Silicon oil was applied around the plates to avoid drying of the sample.

### 2.6. Solubility Analysis in Non-Reducing and Reducing Solutions

The samples, including gels, fillet mince, and protein isolate paste, were ground into a fine paste using an Omni PDH Programmable Digital Homogenizer (Omni International, Kennesaw, GA, USA). For each sample, about 1 g of the fine paste was analyzed for protein content using the Kjeldahl method [13] and this protein content was used to calculate the total protein content of the paste dissolved in non-reducing and reducing solutions. For each sample, about 1 g of the fine paste was weighed into a 50 mL centrifuge tube and then 10 mL of non-reducing solution (5% SDS) or reducing solution (5% SDS + 50 mM β-mercaptoethanol) was added. After homogenization for 1 min at 10,000 rpm, the tube was shaken continuously for 1 h in an orbital shaker. After centrifugation at 14,000× *g* for 20 min, the supernatant was analyzed for soluble protein content using the Kjeldahl method [13]. The solubility in the two extraction solutions was expressed as the percentage of soluble protein relative to the total protein.

### 2.7. Sodium Dodecyl Sulfate-Polyacrylamide Gel Electrophoresis (SDS-PAGE)

In order to better understand the role of disulfide bonds in the gel formation, non-reducing and reducing SDS-PAGEs were conducted according to Laemmli [14]. The protein was extracted as described in the Section 2.6. After mixing with non-reducing sample buffer or reducing sample buffer, 40 µL of sample solution was loaded. To better understand the gelling effect of disulfide bonds and MTGase-induced cross-linking, the protein content loaded was the protein which could be extracted from 40 µg of protein of the original samples based on measurements by the Kjeldahl method. For better separation, a 4% stacking gel and a 4–20% separating gel were employed. Protein marker containing 12 protein standards with molecular weights, ranging from 6.5 to 200 kDa, was also loaded. After electrophoresis, the gels were stained using 0.25% Coomassie Brilliant Blue R-250 solution. The gels were scanned using the Image Lab (ChemiDoc XRS +, Hercules, CA, USA) after de-staining.

### 2.8. Statistical Analysis

Each treatment was conducted in triplicate. In the analyses of color, texture, and solubility, each prepared sample was assayed twice. For color determination, only gels made from protein isolate were measured with the gel without MTGase added as the control. For texture analysis, gels made from fillet mince using both setting method and direct heating method were included for comparison. In solubility analysis and SDS-PAGE, uncooked fillet mince and protein isolate were included to elucidate the disulfide bonds before heating. The data were treated using the analysis of variance (ANOVA) with the SAS 9.4 package. Significant differences among treatment means were determined using Duncan’s multiple range test (α = 0.05). Data are expressed as means ± SD (n = 3). The Pearson correlation coefficient (r) was determined.

## 3. Results and Discussion

### 3.1. Color Profile of Gels Made with Different Levels of MTGase

As shown in Table 1, the addition of MTGase has no significant effect on the L* value and a* value. Guo et al. [15] also found that the color profile of the gels made from silver carp surimi was not affected by MTGase. However, with the same heating practice as that used in the present study, Benjakul et al. [16] found that when MTGase was added over 0.4 units/g, the L* value of gels made from lizardfish surimi increased slightly and these researchers attributed the effect to light scattering from the maltodextrin in MTGase powder. Zhang et al. [17] also found slight increases in the L* value with the addition of MTGase to longtail southern cod mince, but whiteness was not changed. However, as shown in Table 1, the addition of MTGase caused an increase in the b* value from 11.47 to over 12. The MTGase-induced increase in b* was also observed by other researchers [5,16]. The whiteness of gels made from all treatments did not show any significant differences, which was in agreement with the reports of gels made from threadfin bream surimi and alkali-extracted protein from Atlantic menhaden [18,19]. However, some researchers found that MTGase addition could cause a slight whiteness increase in gels made from threadfin bream, Indian mackerel, sardine, and hairtail surimi [2,20]. According to Chanarat and Benjakul [8], the effect of MTGase on gel whiteness was related to the protein arrangement in the network. Given the data presented in the current study and from other researchers, it might be concluded that the application of MTGase has no or slight effects on the color profile of gels made from fish mince and protein isolate. Table 1 also shows that the two heating methods had no different effects on L*, a*, and whiteness, but the gels made using the setting method showed a lower b* value than the gels made using the direct heating method.

### 3.2. Texture Profile Analysis (TPA) of Gels

As shown in Table 2, the setting step significantly improved the textural profile of the gels made from fillet mince. For example, chewiness was increased by about 34% if the setting step was employed. The enhancement of gel strength came from the cross-linking mediated by the endogenous transglutaminases, which was substantiated by the following SDS-PAGE pattern (Figure 1 and Figure 2). However, in the absence of MTGase, the setting step had a negative effect on the texture of gels made from protein isolate. As shown in Table 2, compared with the direct heating method, the setting method caused an 11% decrease in chewiness and a 17% decrease in deformation. The setting-induced texture deterioration stemmed from the proteolysis by endogenous proteases. The highly denatured protein isolate made it easily accessible to the endogenous proteases. Our recently published data revealed that after gelation of the protein isolate with the setting method, the TCA-soluble peptides increased by 140% [10]. The peptides from proteolysis could be visualized in the SDS-PAGE (Figure 1 and Figure 2). With the increase in added MTGase, all textural parameters were substantially increased. With proper amounts of MTGase applied, the gels made from protein isolate could achieve higher hardness and equal springiness (elasticity) in comparison with the gels made from fillet mince. The pronounced improvement in the textural properties had been reported to be due to the dissociation of the protein complex during alkaline protein extraction, which exposed more reactive groups to facilitate cross-linking [8,21]. It should be noted that the cohesiveness could be increased by 54% from 0.28 to 0.43, which was still much lower than that of fillet-mince-made gels. Similarly, with the increase in MTGase added, deformation, an indicator of cohesiveness and elasticity, increased from 6.51 to 11.13 mm for the gels made from protein isolate. However, those gels made from fillet mince, either with or without the setting step, did not fracture under 75% compression. For the alkali-extracted byproduct protein isolate, the MTGase-induced cross-linking seemed more effective to increase hardness than cohesiveness. As shown in Table 2, when MTGase reached 0.5 units/g, the hardness increased by 218%, but the cohesiveness and deformation only increased by 55 and 71%, respectively. In comparison with the gels made from the protein isolate without MTGase added, the gels made with 0.5 units/g showed a 218, 55, 12, 451, 115, 446, and 71% increase in hardness, cohesiveness, springiness, chewiness, resilience, fracturablity, and deformation, respectively.

It should be noted that with the increase in added MTGase up to 0.5 units/g, all textural parameters increased in a dose-dependent manner. However, a further increase to 0.6 units/g had almost no effect on the texture profile. This was different from our previous study [5], which showed that when the protein isolate was extracted at pH 8.5 from catfish byproducts, the breaking force and deformation of gels could increase with the increase in MTGase up to 1.0 units/g. This difference should be due to the different denaturation states of the protein isolates as the highly unfolded structure and exposed active groups of pH-11-extracted protein isolate rendered it a better substrate for MTGase. Therefore, there is no universal recipe for the MTGase addition in terms of how many enzyme units are needed to induce the optimal cross-linking effect. The use of enzyme activity for a given material was determined by the specific structural characteristics of proteins and other factors during gel making. Our results help to explain the discrepancies reported in the literature in that few researchers explained their results with the status of protein structures in their samples/raw materials. Asagami et al. [22] found that the textural properties of gels made from the undenatured proteins of seven different fish species showed inconsistent results with the increase in MTGase from 0 to 1.1 units/g. These researchers found that a high level of added MTGase could lead to a decrease in punch force and, particularly, punch deformation. Chanarat et al. [2] found that 0.2 units/g could achieve the highest breaking force and deformation for threadfin bream-made gels and beyond 0.2 units/g marked decreases in these two values were observed. Meanwhile, for Indian mackerel and sardine, the two parameters could increase until MTGase level reached 0.6 units/g. Hu et al. [20] found that the breaking force and deformation of the gels made from hairtail surimi increased with MTGase increased up to 0.4 units/g, after which a slight decrease in both parameters was observed. Seguro et al. [23] even found that gels made from different grades of Alaska Pollock surimi exhibited pronounced differences in their response to MTGase. With the increase in added MTGase, the deformation of the gels made from SA grade Alaska Pollock surimi decreased steadily, while the gels made from second grade Alaska Pollock surimi showed an opposite trend. These researchers concluded that there was a critical degree of ε-(γ-Glu)Lys cross-linking, beyond which no or even an impairing effect of MTGase on the gel strength could ensue. The detrimental effect on gel strength of the high level of MTGase could be attributed to the self-aggregation of MTGase and rapid and excessive polymerization of myofibrillar proteins, which compromised the ordered gel matrix [2,20].

Except fish species, the protein conformation or the state of the protein structure can affect the reactivity of fish protein to MTGase. For the gels made from frozen-stored longtail southern cod, Zhang et al. [17] found that when MTGase was added above 0.2 units/g, breaking force, hardness, springiness, chewiness, and resilience plateaued. Using a mixture of protein isolates extracted from eviscerated silver carp and kilka at pH 11.5, Abdollahi et al. [7] reported that the breaking force and deformation of the gels could reach a maximum at only 0.2 units/g. On the contrary, Chanarat and Benjakul [8] even found that gels made from Indian mackerel protein isolate showed higher deformation than gels made from unwashed mince. This pronounced disparity between our current study and above-mentioned reports could be attributed mostly to a higher degree of denaturation of the catfish byproduct protein isolate, which was extracted at pH 11 in our study. As revealed in our previous report [6], after 45 min of alkaline protein extraction, protein isolate was highly denatured as evidenced by lower salt solubility and greatly changed secondary structures. By contrast, the frozen fish and alkali-extracted isolates reported by others as mentioned above might not have been as heavily denatured as the catfish protein isolate used in our current study. In view of the effects of fish species and protein conformation on the response to MTGase, it is hard to make a comparison with the literature, since protein structure had not been reported in these studies. Therefore, in the current study, the gels made from protein isolate were compared with gels made from catfish fillet.

### 3.3. Dynamic Rheological Analysis

As shown in the rheogram (Figure 3), all catfish isolates, either with MTGase or not, showed a similar gelation pattern, which suggested that the addition of MTGase only improved the gel strength but did not change the gelation mechanism. From the beginning, the storage modulus (G’) deceased until 30 °C, and then increased slightly to 35 °C. From 35 to 40 °C, especially at 40 °C, a sharp increase occurred. This increase might stem from the hydrophobic interactions, as the fillet mince also showed a small increase from 35 to 40 °C. In DSC analysis (differential scanning calorimetry), Raghavan and Kristinsson [3] found that the myosin of catfish showed two transition peaks around 40 °C. From 40 °C, G’ exhibited another drop until 50 °C. This drop might be caused by protein degradation induced by endogenous enzymes. The rheogram clearly shows different gelation patterns between the protein isolate and fillet mince, which could be largely attributed to their different protein conformation and enzyme composition. For example, fillet mince showed a sharp decrease from 40 to 55 °C (Figure 3), but protein isolate only showed a steady decrease until 50 °C. This difference might be due to the higher level of proteolytic enzymes in the fillet mince [24]. Figure 3 exhibited totally different patterns from the rheograms acquired from our previous study (Figure 2 of our 2021 report [5]). The G’ storage module force for our current pH-11-extracted proteins towards the gelling temperatures (70–90 °C) was in a much higher force range (>20,000 Pa) than that reported in the pH 8.5 protein gelling patterns (<20,000 Pa). The primary reason for these discrepancies was due to the differences in protein structures between the two different protein samples isolated at different pH values. Figure 4 shows the change in the storage modulus without the setting step. This figure shows a completely different pattern from Figure 3, which indicated that the heating methods had a vital effect on the gelation process and, therefore, the texture of the resultant gels. The TPA data presented in Table 2 substantiated the great effects of the two heating methods on the textural profile of gels.

### 3.4. Solubility in Non-Reducing and Reducing Solutions

In our recent study, the non-reducing and reducing SDS-PAGE revealed that the two major myofibrillar protein components, myosin and actin, were actively involved in the disulfide bond formation during the gelation process due to their denatured state [10]. The protein solubility in non-reducing and reducing solutions was another measure of the role of disulfide bonds in the gel formation. For fillet mince, protein solubility in non-reducing and reducing solutions was 90.63 and 92.6%, respectively (Table 3). The lower solubility than the protein isolate should be partially due to some connective tissue proteins such as stroma proteins [16]. It was obvious that fillet mince did not show any significant differences under non-reducing and reducing conditions, indicating that the fresh fillet mince lacked process-induced disulfide bonds, which may promote the formation of the insoluble aggregates. In general, for protein isolates, the solubility, either in non-reducing or reducing solution, decreased with the increase in added transglutaminase. This was reasonable since more MTGase-induced cross-linking was formed. For the protein isolate gels made without MTGase added, solubility in reducing solution was 23% higher than that in non-reducing solution. Those gels made with MTGase showed a 42–55% solubility increase in reducing solution in comparison with non-reducing solution. The above marked solubility differences between non-reducing and reducing solutions indicated the vital role of disulfide bonds in the gelation of protein isolates, even in the presence of transglutaminase. The much higher disulfide bond formation in the protein isolate than in the fillet mince should be due to the exposed sulfhydryl groups of the denatured protein isolate. Abdollahi et al. [7] found that the protein isolate derived from kilka fish at pH 11.5 had around a 6% higher reactive sulfhydryl content than its fillet mince counterpart. This was also in agreement with the electrophoretic profile. As shown in Figure 1 and Figure 2, the MHC bands (>200 kDa) for gels made from protein isolate did not show up in the non-reducing SDS-PAGE; however, in the reducing SDS-PAGE intense MHC bands appeared. The solubility of protein isolate paste in non-reducing solution before heating was 93.48% and it increased to 96.95% in reducing solution. This phenomenon clearly indicated that disulfide bonds formed even prior to heating. This should be due to the exposed sulfhydryl groups of the denatured protein. The formation of disulfide bonds in protein isolate paste prior to heating was evidenced in the SDS-PAGE results (Lane 4 in Figure 1 and Figure 2). In the non-reducing SDS-PAGE, there are multiple high-molecular-weight bands above MHC even at the top of the stacking gel. However, in the reducing SDS-PAGE with β-mercaptoethanol present, all of these bands disappeared with a concomitant intensity increase in the MHC band. Gelleland et al. [21] found that when Alaska pollock surimi was treated at 300 MPa for 5 min, the solubility of the surimi or surimi-made gels was 42–91% lower in SDS–urea solution than in SDS–urea–β-mercaptoethanol solution. Cardoso et al. [25] found that the gels made from hake fillet mince showed about 10% lower solubility in a SDS solution than in a SDS + DTT (dithiothreitol, a reducing agent) solution. The above differences in percentage solubility changes might stem from distinct material, gelation method, and protein analysis methods. However, all studies, including our current research, clearly demonstrated that disulfide bond formation played a vital role in the gelation process, especially in the proteins with a denatured state. In our current study, the solubility of the fillet mince and the gels made from it did not show any significant differences in reducing solution. However, Hu et al. [20] found that the gels made from hairtail surimi showed a lower solubility in the reducing buffer (1% SDS + 8 M urea + 2% β-ME) than the surimi and attributed it to the formation of non-disulfide bonds mediated by the endogenous TGase. This phenomenon should be attributed to the endogenous protease-induced protein degradation, which cancelled out the effect of TGase. The protease-induced protein degradations in fillet mince during the gelation process can be visualized using the newly formed low-molecular-weight bands in Figure 1. As shown in Table 3, in the reducing solution, all MTGase-added gels showed 87.49–95.35% solubility. This high solubility suggested that a low degree of cross-linking was formed. Whether the low degree of cross-linking is related to the high degrees of disulfide bond formation remains to be further studied. It should be noted that the solubility in non-reducing and reducing solutions only gave an estimate of the role disulfide bonds played in the gelation process. However, other forces, such as hydrophobic interaction, should also facilitate the gel formation in the highly denatured protein isolate. Therefore, more studies are required to elucidate the forces involved in the gelation process of a highly denatured protein system.

### 3.5. SDS-PAGE Patterns of Proteins and Gels

For fillet mince without heating (Lane 1, Figure 1), in non-reducing SDS-PAGE, there were only two small bands in the stacking gel (one within the stacking gel and the other on the top of the stacking gel) above the dense MHC band (200 kDa). After heating, either without setting (Lane 2) or with setting (Lane 3), the lanes above MHC became smeared with the two bands becoming much denser. Concomitantly, the MHC band became less intense. This clearly indicated that high-molecular-weight protein aggregates were formed during the gelation process. By contrast, in reducing SDS-PAGE (Figure 2), most of these smears and dense bands disappeared except one dense band on the top of the stacking gel for the gel made from fillet mince using the setting step (Lane 3). This band should be a cross-linked protein aggregate induced by endogenous transglutaminase in the fillet mince, because it could not be disassociated using β-mercaptoethanol. The band above the stacking gel in non-reducing SDS-PAGE (Lane 2) was not present in the reducing SDS-PAGE, indicating that this band was an aggregate formed by disulfide bonds and the endogenous transglutaminase in catfish could not form cross-linking if a setting step (such as setting at 40–45 °C for a period of time before increasing the temperature) was not adopted. This phenomenon was consistent with the solubility data presented in Table 3, in which the solubility of the gel made without a setting step increased from 88.71% in non-reducing solution to 94.68% in reducing solution. As shown in Figure 4, in the direct heating (without a setting step), the temperature of the paste was increased quickly to above 50 °C within 8 min. This fast heating process inhibited the action of endogenous transglutaminase. However, the MHC bands in the fish protein gels in the reducing SDS-PAGE (Lane 2 and 3) showed markedly increased intensity in comparison with non-reducing SDS-PAGE. The above observation clearly manifested that disulfide bonds contributed significantly to the gel formation in fillet mince protein.

For the protein isolate (Lane 4) in non-reducing SDS-PAGE, a dense smear above the MHC band spreading all the way to the upper edge of the stacking gel and a dense band on the top of the stacking gel were observed. However, in reducing SDS-PAGE, all the smear and dense bands vanished with the substantially increased intensity of MCH. This indicated that, even if without heating, disulfide bonds could also form in the highly denatured protein isolate. The gels made from protein isolate without (Lane 5) and with (lane 6) a setting step also showed a similar trend except that the latter (Lane 6) had a faint MHC band due to the protein degradation caused by endogenous proteases. However, in response to the much lower intensity of MCH in Lane 6 in both non-reducing SDS-PAGE and reducing SDS-PAGE, multiple low-molecular-weight bands appeared in this lane, particularly those intense bands enclosed in circles. In addition, the lower part of Lane 6 became heavily smeared. This further revealed the high activity of endogenous proteases present in the protein isolate. These newly emerging bands should be derived from MHC breakdown [26]. In our recently published study, it was found that the TCA-soluble peptide content of protein isolate could be increased by 132% after gelation [10]. This proved the high activity of endogenous proteases during the heating process, especially during the setting step. This also explained the poorer textural profile of the gels made with setting than that without setting (Table 2).

For those gels made with the addition of transglutaminase (Lanes 7–12), in non-reducing SDS-PAGE, no bands including MCH were resolved in the upper part of the separating gel, but one dense band in the stacking gel and one even denser band on the top of the stacking gel were observed. In reducing SDS-PAGE, multiple bands including MCH showed up. This demonstrated that some proteins including myosin, were not totally involved in the transglutaminase-induced cross-linking. A portion of these proteins was only linked by disulfide bonds and was released when β-mercaptoethanol was present. However, it was also observed that in the reducing SDS-PAGE, if no MTGase was applied (Lane 6), the bands in the stacking gel vanished. On the other hand, if MTGase was applied, the bands on the top of the stacking gel still existed (lane 7–12), indicating that these bands were protein aggregates formed mostly by transglutaminase.

Compared with fillet mince (Lane 1), protein isolate (Lane 4) showed some new bands (enclosed in rectangles). It was not clear whether these new bands were derived from alkali-caused hydrolysis or alkaline protease-induced proteolysis during protein extraction. On the contrary, some bands in fillet mince as marked by arrows (Lane 1) became faint or disappeared in the protein isolate (Lane 4). For example, a band measured 51 kDa appeared in the fillet mince (Lane 1) but was not detected in the protein isolate. Marmon and Undeland [27] also found the disappearance of this band in the alkali-extracted protein isolate at pH 11.2 from herring and tentatively identified it as desmin. These disappearing bands should be sarcoplasmic proteins, which were removed in the washing water or in the supernatant after acidic precipitation. Electrophoresis of proteins in washing water and supernatant should be conducted to substantiate this hypothesis.

Unlike myosin, the actin band showed no visible changes in response to different levels of MTGase added. The resistance of actin to the MTGase-induced cross-linking had been reported in various fish species [8,28]. According to Ramirez-Suarez et al. [29], actin was not easily cross-linked due to the lack of exposed glutamine and lysine. However, with purified rabbit proteins, Huang et al. [30] found that actin could form cross-links with HMM or with itself, even though these cross-links were formed slowly. Our recently published DSC data revealed that the addition of MTGase to catfish protein isolate did not change the maximum temperature (*T*_max_) of actin, indicating little or no involvement of actin in the MTGase-induced cross-linking. According to Seguro et al. [23], cross-linking increased with the increase in MTGase applied, even though the gel strength was not improved accordingly. However, differently from other reports [8], the intensity of MHC band did not exhibit a definite dose-dependent decrease with the increase in MTGase applied (Figure 4). This should be due to enzyme-induced proteolysis, which was greatly affected by MTGase. As revealed in our recently published paper [10], the endogenous enzymes in the protein isolate were still very active and led to the proteolysis of proteins, especially myosin. In response to its lower MHC intensity, the gel made without MTGase (Lane 6) showed more small bands and more smeared low-molecular-weight regions than those gels made with MTGase (Lane 7–12). In addition, in both non-reducing and reducing SDS-PAGE, the low-molecular-weight area became less smeared with the increase in MTGase added. This phenomenon implies that MTGase might exert an inhibitory effect on the proteolysis. The compact and ordered gel microstructure caused by cross-linking might limit the access of endogenous enzymes [28,31]. Fang et al. [31] found that when MTGase was added, the gel made from silver carp surimi showed significantly lower digestibility by pepsin. Yongsawatigul and Piyadhammaviboon [32] found that, with the increased level of MTGase added into lizardfish surimi, the TCA-soluble oligopeptides content decreased and attributed it to the resistance of ε-(Υ-glutamyl)lysyl isopeptides to the endogenous proteases. Yerlikaya et al. [33] also reported that the addition of MTGase into rainbow trout mince suppressed the activity of enzymes. Our recently published study found that the addition of MTGase to the protein isolate derived from catfish byproducts increased the enthalpy (Δ*H*) and maximum temperature (*T*_max_) of myosin, which demonstrated the conformational change of myosin by MTGase [5]. In addition to this, we speculate that the added MTGase might also cause the cross-linking of the endogenous proteases and, as a result, weaken their proteolytic activity to some extent. The fact that MHC bands did not show decreased intensity with the increase in added MTGase might be due to the compromised activity of endogenous proteases. Differently from our previous study [6] and those reported in the literature, our current study employed both non-reducing and reducing SDS-PAGE. The comparison of the two electrophoretic patterns provided insight into how the disulfide bonds function in the gelation process, especially when non-denatured fillet mince and highly denatured protein isolate were compared.

### 3.6. Pearson Correlation Analysis among Parameters

As shown in Table 4, non-reducing solubility and reducing solubility only exhibited a moderate correlation (r = 0.759). This proved the above-mentioned speculation that there should be some interactions between disulfide bond formation and MTGase-induced cross-linking during the gelation process. Hardness had a moderate correlation with G’ (r = 0.839). Even though these two parameters are both indicators of firmness, they are differently measured, based on normal force and shear stress, respectively. Therefore, the current study depicted the gels from textural and rheological perspectives and produced more comprehensive information. Cohesiveness, a measure of how well the product withstands a second deformation relative to its resistance under the first deformation, was strongly correlated to deformation (r = 0.967) and resilience (r = 0.977).

## 4. Conclusions

MTGase proved to be an effective agent to improve the texture of gel made from protein isolate extracted from catfish byproducts. However, there existed a critical MTGase level of 0.5 units/g, beyond which further increases in enzyme units had no or even negative effects on the texture profile. Compared with the gels made from fillet mince, the gels made from protein isolate from alkaline-extracted byproducts still lacked cohesiveness. Disulfide bonds played an important role in the gelation of protein isolate. The new bands appearing in the SDS-PAGE after gelation of protein isolate indicated endogenous enzyme-induced proteolysis even in the presence of MTGase. The highly denatured state of protein isolate made it behave differently in the gelation process in comparison with the catfish fillet. The different gelation patterns exhibited by the protein isolate and fillet mince suggested that different gelation mechanisms were involved. In view of the susceptibility of protein isolate to the endogenous enzymes during gelation, enzyme inhibitors should be employed in combination with MTGase to further enhance the gel texture.

## Figures and Tables

**Figure 1 foods-12-02029-f001:**
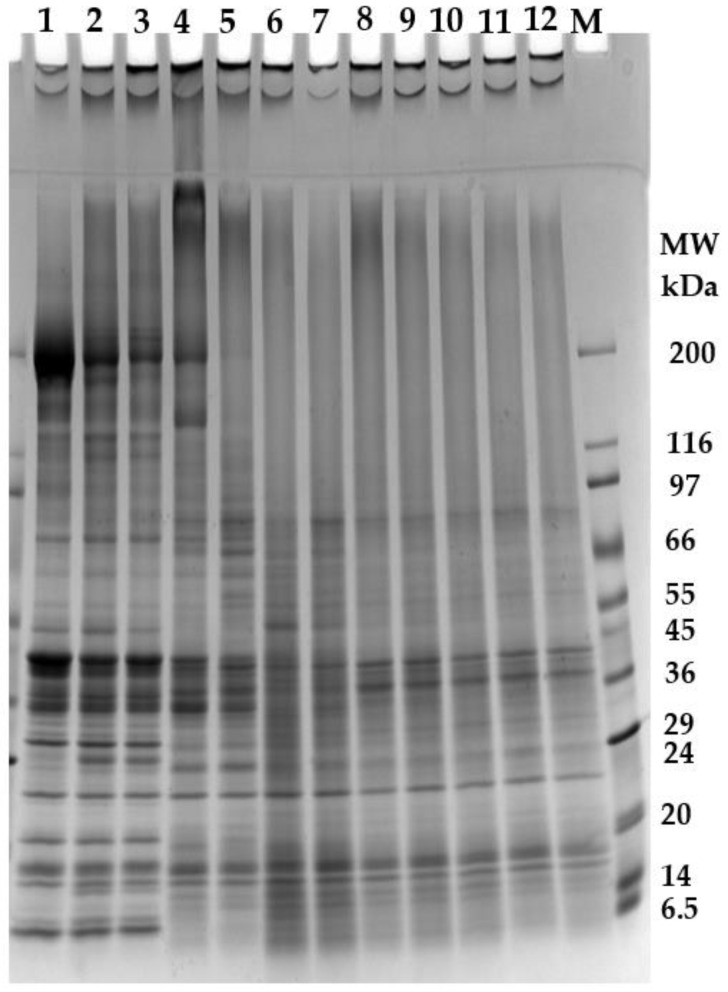
Non-reducing SDS-PAGE. Lane M: molecular weight marker (6.5–200 kDa); Lane 1: catfish fillet mince; Lane 2: gel made from fillet mince without setting; Lane 3: gel made from fillet mince with setting; Lane 4: protein isolate; Lane 5: gel made from protein isolate without setting; Lane 6: gel made from protein isolate with setting; Lanes 7–12: gels made from protein isolate with 0.1, 0.2, 0.3, 0.4, 0.5, and 0.6 units of MTGase/g, respectively.

**Figure 2 foods-12-02029-f002:**
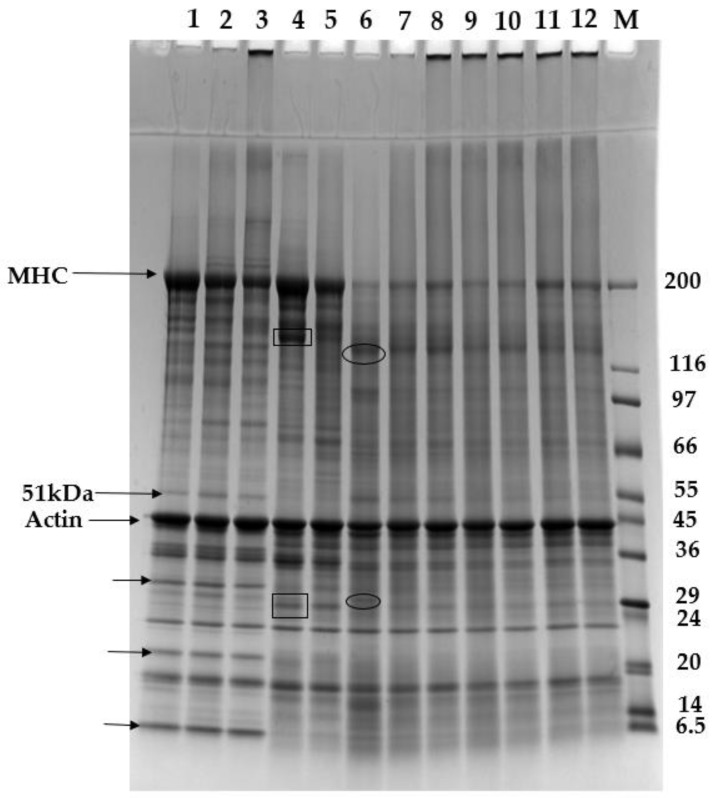
Reducing SDS-PAGE. Lane M: molecular weight marker (6.5–200 kDa); Lane 1: catfish fillet mince; Lane 2: gel made from fillet mince without setting; Lane 3: gel made from fillet mince with setting; Lane 4: protein isolate; Lane 5: gel made from protein isolate without setting; Lane 6: gel made from protein isolate with setting; Lanes 7–12: gels made from protein isolate with 0.1, 0.2, 0.3, 0.4, 0.5, and 0.6 units of MTGase/g, respectively.

**Figure 3 foods-12-02029-f003:**
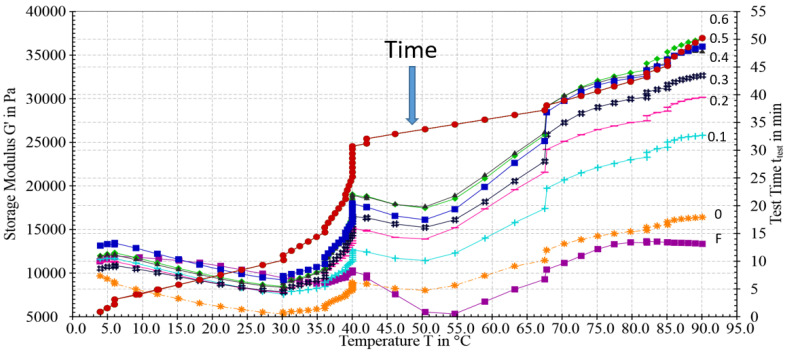
Change in storage modulus (G’) during sol–gel transition mimicking the setting method (40 °C for 30 min followed by 90 °C for 20 min). F denotes fillet mince; 0, 0.1, 0.2, 0.3, 0.4, 0.5, and 0.6 denote the protein isolates added with different concentrations of MTGase.

**Figure 4 foods-12-02029-f004:**
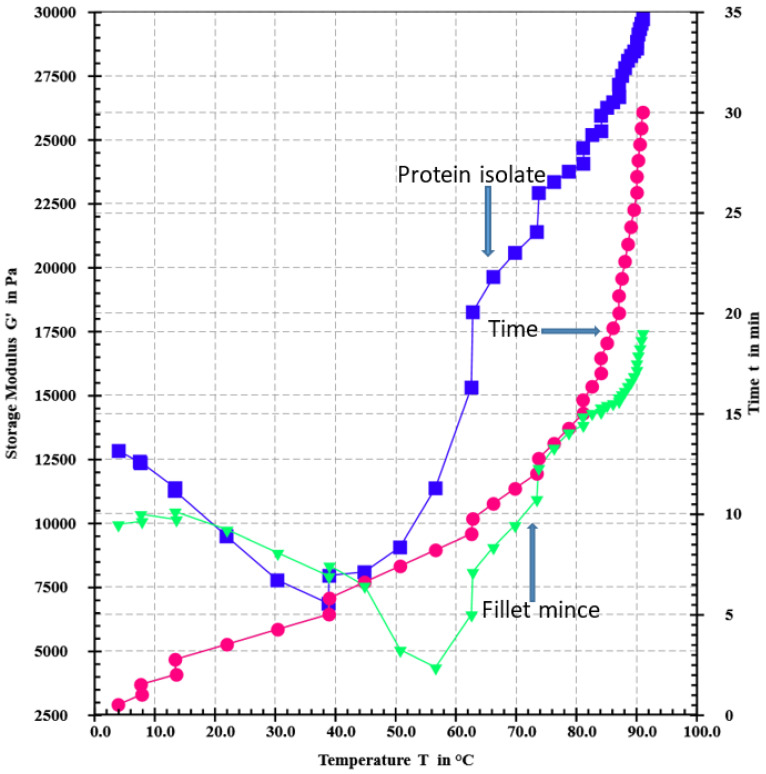
Change in storage modulus (G’) during sol–gel transition mimicking the direct heating method (90 °C for 30 min).

**Table 1 foods-12-02029-t001:** Color profile of gels made from protein isolate with different levels of MTGase.

Samples	L*	a*	b*	Whiteness
Protein isolate without setting	74.17 ± 0.74 a	−1.04 ± 0.13 a	12.26 ± 0.27 a	71.39 ± 0.74 a
Protein isolate with setting	74.77 ± 1.10 a	−1.07 ± 0.09 a	11.47 ± 0.31 b	72.26 ± 0.81 a
0.1 units/g with setting	74.72 ± 0.36 a	−0.97 ± 0.10 a	12.17 ± 0.24 a	71.93 ± 0.44 a
0.2 units/g with setting	74.20 ± 0.54 a	−1.01 ± 0.11 a	12.17 ± 0.30 a	71.46 ± 0.49 a
0.3 units/g with setting	74.40 ± 0.49 a	−0.95 ± 0.02 a	12.32 ± 0.17 a	71.57 ± 0.27 a
0.4 units/g with setting	74.11 ± 0.82 a	−0.99 ± 0.06 a	12.18 ± 0.21 a	71.37 ± 0.78 a
0.5 units/g with setting	74.76 ± 1.33 a	−1.02 ± 0.05 a	12.31 ± 0.29 a	71.90 ± 0.91 a
0.6 units/g with setting	74.99 ± 1.47 a	−1.07 ± 0.08 a	12.40 ± 0.25 a	72.06 ± 0.40 a

Values in the same column with the same letter are not significantly different (*p* < 0.05).

**Table 2 foods-12-02029-t002:** Texture profile analysis of gels made with different heating methods and different levels of MTGase.

Samples	Hardness at Maximum Force (g)	Cohesiveness	Springiness (%)	Chewiness (g)	Resilience (%)	Fracturability (g)	Deformation (mm)
Fillet mince without setting	22,504 ± 1328 c	0.626 ± 0.005 a	80.11 ± 1.04 b	11,278 ± 502 c	26.08 ± 0.41 b	ND	ND
Fillet mince with setting	27,096 ± 762 b	0.648 ± 0.014 a	85.77 ± 1.43 a	15,058 ± 181 a	29.93 ± 0.57 a	ND	ND
Protein isolate without setting	10,288 ± 1032 f	0.303 ± 0.011 de	79.98 ± 2.08 b	2484 ± 31 g	8.09 ± 0.11 f	8963 ± 609 e	7.88 ± 0.27 e
Protein isolate with setting	9975 ± 366 f	0.280 ± 0.005 e	79.03 ± 0.52 b	2203 ± 27 g	7.27 ± 0.38 f	5823 ± 343 f	6.51 ± 0.19 f
0.1 units/g with setting	11,335 ± 577 f	0.318 ± 0.012 d	80.34 ± 2.88 b	2890 ± 144 g	8.19 ± 0.06 f	9613 ± 267 e	8.21 ± 0.16 d
0.2 unit/g with setting	15,243 ± 619 e	0.355 ± 0.019 c	85.80 ± 0.17 a	4633 ± 139 f	9.65 ± 0.29 e	14,258 ± 332 d	9.83 ± 0.15 c
0.3 units/g with setting	16,993 ± 82 d	0.362 ± 0.004 c	87.62 ± 0.32 a	5382 ± 59 e	10.30 ± 0.23 e	16,993 ± 82 c	9.80 ± 0.06 c
0.4 units/g with setting	23,199 ± 856 c	0.416 ± 0.026 b	86.17 ± 1.47 a	8304 ± 314 d	12.95 ± 0.70 d	23,199 ± 856 b	10.75 ± 0.13 b
0.5 units/g with setting	31,768 ± 520 a	0.434 ± 0.025 b	88.17 ± 1.45 a	12,145 ± 700 b	15.61 ± 0.92 c	31,768 ± 520 a	11.13 ± 0.06 a
0.6 units/g with setting	30,638 ± 615 a	0.433 ± 0.022 b	87.81 ± 1.39 a	11,635 ± 639 bc	15.23 ± 1.79 c	30,938 ± 191 a	11.02 ± 0.15 ab

Values in the same column with the same letter are not significantly different (*p* < 0.05). ND: fracture was not detected at 75% compression.

**Table 3 foods-12-02029-t003:** Solubility in non-reducing and reducing solutions.

	Protein Solubility (%)	
Sample	Non-Reducing (5% SDS)	Reducing (5% SDS + 50 mM 2-ME)
Fillet mince	90.63 ± 0.50 aA	92.60 ± 1.53 cdeA
Gel made from fillet mince without setting	88.71 ± 0.37 abB	94.68 ± 0.39 abcdA
Gel made from fillet mince with setting	96.36 ± 0.12 aA	96.66 ± 1.29 abcA
Protein isolate	93.48 ± 0.47 aB	96.95 ± 0.1 abA
Gel made from protein isolate without setting	79.66 ± 7.61 bcB	97.91 ± 2.95 aA
Gel made from protein isolate with setting	77.28 ± 4.90 cB	95.36 ± 1.68 abcdA
Gel made from protein isolate with 0.1 unit MTGase/g	65.64 ± 1.08 dB	93.41 ± 1.96 bcdA
Gel made from protein isolate with 0.2 unit MTGase/g	66.16 ± 8.76 dB	95.25 ± 1.21 abcdA
Gel made from protein isolate with 0.3 unit MTGase/g	63.08 ± 9.04 dB	95.35 ± 1.68 abcdA
Gel made from protein isolate with 0.4 unit MTGase/g	60.79 ± 1.20 dB	91.73 ± 2.76 deA
Gel made from protein isolate with 0.5 unit MTGase/g	57.46 ± 1.89 dB	89.07 ± 0.67 efA
Gel made from protein isolate with 0.6 unit MTGase/g	57.58 ± 1.37 dB	87.49 ± 1.61 fA

Values in the same column with the same lowercase letter are not significantly different (*p* < 0.05). Values in the same row with the same capital letter are not significantly different (*p* < 0.05).

**Table 4 foods-12-02029-t004:** Pearson correlation analysis among parameters.

	Reducing Solubility	Hardness	G’	Cohesiveness	Springiness	Chewiness	Resilience	Fracturability	Deformation
Non-reducing solubility	0.759 *	−0.858 *	−0.971 ***	−0.935 **	−0.849 *	−0.860 *	−0.882 **	−0.902 **	−0.949 **
Reducing solubility		−0.904 **	−0.666	−0.817 *	−0.518	−0.905 **	−0.895 **	−0.881 **	−0.676
Hardness			0.839 *	0.956 ***	0.803 *	0.999 ***	0.998 ***	0.995 ***	0.868 *
G’				0.950 **	0.929 **	0.841 *	0.869 *	0.890 **	0.989 ***
Cohesiveness					0.882 **	0.957 ***	0.973 ***	0.977 ***	0.967 ***
Springiness						0.801 *	0.819 *	0.848 *	0.940 **
Chewiness							0.998 ***	0.995 ***	0.870 *
Resilience								0.998 ***	0.894 **
Fracturability									0.911 **

G’ is the storage modulus at the end of heating with the setting method. * means correlation at α = 0.05. ** means correlation at α = 0.01. *** means correlation at α = 0.001.

## Data Availability

The datasets generated for this study are available upon request to the corresponding author.

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
