# Peer review of "Microbial Transglutaminase Cross-Linking Enhances the Textural and Rheological Properties of the Surimi-like Gels Made from Alkali-Extracted Protein Isolate from Catfish Byproducts and the Role of Disulfide Bonds in Gelling"

_foods, 2023, doi:10.3390/foods12102029_

Round 1

Reviewer 1 Report

这是一份好工作。为提高鲶鱼副产物的利用率,本研究在从鲶鱼中提取的蛋白质分离物中加入不同水平的0.1-0.6 unit/g微生物谷氨酰胺转移酶(MTGase),分析其凝胶的质构特征,揭示二硫键在凝胶上MTGase机制中的作用

Author Response

Response to reviewer 1

Thank you for your comments.

Reviewer 2 Report

General comments:

In this study, different levels of microbial transglutaminase (MTGase) were applied to improve the textural properties of gels prepared from protein isolate extracted from catfish byproducts. To its credit, the article is clearly structured. However, this article has some problems that must be revised.

Specific comments:

1.               The abstract needs to be revised, and the abstract should reflect the conclusions and ideas of the entire article, including the purpose, method, results and application of the research work.

2.               The keywords are meaningless, please revised.

3.               Whether there is a determination of the water holding capacity of gels, please add.

4.               Whether there are pictures of gel samples and the microstructure of gels, please add.

5.               It is not comprehensive if only disulfide bonds are to be discussed to explain the interaction forces during gel formation, please consider it again.

6.               The conclusion needs to be revised, and the conclusion should reasonably answer the questions raised in the introduction.

7.               Article Title “Microbial transglutaminase enhances the gelation properties of protein isolate extracted from catfish byproducts” does not fully summarize the main purpose of your article, please consider it again.

Author Response

Dear Editor(s) and reviewer: Thank you very much for your kindness in pointing out errors or corrections or considerations for improving the quality of the manuscript. We have made revisions according to the comments of reviewers.

Response to reviewer 2

General comments: In this study, different levels of microbial transglutaminase (MTGase) were applied to improve the textural properties of gels prepared from protein isolate extracted from catfish byproducts. To its credit, the article is clearly structured. However, this article has some problems that must be revised.

Response: Problems have been addressed.

Comment 1:   The abstract needs to be revised, and the abstract should reflect the conclusions and ideas of the entire article, including the purpose, method, results and application of the research work.

Response: The abstract has been revised accordingly (lines 13-33).

Comment 2:  The keywords are meaningless, please revised.

Response: Some keywords have been changed (Lines 34-35)

Comment 3: Whether there is a determination of the water holding capacity of gels, please add.

Response: Thank you for your good suggestion. However, the water holding capacity was not measured in the current study and will be included in the following studies.

Comment 4:  Whether there are pictures of gel samples and the microstructure of gels, please add.

Response: Thank you for your valuable suggestion. In current study, no pictures were taken. Scanning electron microscopy will be conducted in the future studies.    

Comment 5: It is not comprehensive if only disulfide bonds are to be discussed to explain the interaction forces during gel formation, please consider it again.

Response: The reviewer’s point is correct. Except disulfide bonds, during gelation, other forces especially hydrophobic interaction, also contribute to the gel formation. In further studies, other forces involved in the gelation will be investigated. A comment was given in the text (Lines 375-380).

Comment 6: The conclusion needs to be revised, and the conclusion should reasonably answer the questions raised in the introduction.

Response: Conclusion has been revised accordingly (Lines 524-537).

Comment 7: Article Title “Microbial transglutaminase enhances the gelation properties of protein isolate extracted from catfish byproducts” does not fully summarize the main purpose of

Response: We have changed the title to:

Microbial transglutaminase cross-linking enhances the textural and rheological properties of the surimi-like gels made from alkali-extracted protein isolate from catfish byproducts and the role of disulfide bonds in gelling (Lines 2-5)

This is a bit long, but would reflect better of our objectives.

Reviewer 3 Report

Line 98-100: It had been stated that heat treatment is carried out with two different methods.However, the interaction of both the heat treatment method and the use of MTGase had  not been evaluated. Interactions must also be given in the results.

The experimental design is not also clear.

On the other hand, statistical evaluation (line 152) must be written in more detail. Is there a block in the application?

The samples given in tables 1, 2 and 3 were different from each other. Please give an explanation.

Line 490: How were conclusions reached regarding endogenous enzymes? Clearer expressions should be used.

Author Response

Response to reviewer 3

Comment 1: Line 98-100: It had been stated that heat treatment is carried out with two different methods. However, the interaction of both the heat treatment method and the use of MTGase had  not been evaluated. Interactions must also be given in the results.

Response: In this study, when MTGase was added, only setting method was used, because MTGase has high activity at around 40 °C. For comparison, direct heating method was only applied to fillet mince and the protein isolate without MTGase added. To make it clear, I wrote the method (Lines 113-117)

Comment 2: The experimental design is not also clear.

Response: Change has been made to make the experimental design clear (Lines 113-117; Lines 172-175).

Comment 3: On the other hand, statistical evaluation (Line 152) must be written in more detail. Is there a block in the application?

Response: Statistical analysis has been written in detail. Yes, this experiment was conducted in triplicate (Lines 172-175).

Comment 4: The samples given in tables 1, 2 and 3 were different from each other. Please give an explanation.

Response: In Table 1, the main purpose was to determine the effect of MTGase on color profile of gels made from protein isolate. Therefore, gels made form fillet mince, which was used as a reference for comparison and had not been treated with MTGase, were not included. In Table 3, fillet mince and protein isolate without heat-setting were included, because the disulfide bonds need to be investigated in the material before heating.

Comment 5: Line 490: How were conclusions reached regarding endogenous enzymes? Clearer expressions should be used.

Response: A change has been made to get it clear (Lines 529-531).

Reviewer 4 Report

Manuscript title: Microbial transglutaminase enhances the gelation properties of protein isolate extracted from catfish by-products. In this study authors used enzyme to enhance the gelation properties of protein isolate extracted from catfish by-products and its characteristics. The manuscript has very few experiments and is suitable for short communication. As authors claimed to improve the gel forming ability, authors should perform the gelling characteristics of sample in few other experiments

Abstract

Authors should introduce the background of the study, why we need to improve the gel properties.

Objectives are not clear

Recommendations of the research must be added  

Introduction

This section is clear and address the study approach

Methods

2.3. Color analysis of gels: provide the methodology in-detail; what is control here; how authors performed calibration, tile?

2.5. Dynamic rheological analysis: add citation

2.7. Sodium dodecyl sulfate-polyacrylamide gel electrophoresis (SDS-PAGE): add ref

Very weak experimental analysis

No experiments on gel forming ability; GEL STABILITY ETC

No experiment on foaming capacity

This study is suitable for short communication.

Results and Discussion

Since the methodology is very weak, results are not very interesting. Authors used a few experiments and performed correlation analysis. At least authors should perform the PCA and HCA analysis to discriminate the samples with or without addition of enzymes.

Food application of gel is also imp. Authors performed no studies in this section.

Overall, this study is incomplete with fewer experiments.

Author Response

Response to reviewer 4

Comment 1: Manuscript title: Microbial transglutaminase enhances the gelation properties of protein isolate extracted from catfish by-products. In this study authors used enzyme to enhance the gelation properties of protein isolate extracted from catfish by-products and its characteristics. The manuscript has very few experiments and is suitable for short communication. As authors claimed to improve the gel forming ability, authors should perform the gelling characteristics of sample in few other experiments

Response: Thank you for your valuable comment. The main purpose of this study is to solve the texture problem of the gels made from protein isolate. In our previous study, we have found the gels made from protein isolate was brittle and lacked elasticity. As the reviewers suggested, more experiments will be done to further dig into the gelation process.

Comment 2: Abstract

Authors should introduce the background of the study, why we need to improve the gel properties.

Objectives are not clear

Recommendations of the research must be added  

Response: Some changes have been made accordingly (Lines 13-33).

Comment 3: Introduction

This section is clear and address the study approach

Response: Thank you for your comments.

Comment 4: Methods

2.3. Color analysis of gels: provide the methodology in-detail; what is control here; how authors performed calibration, tile?

Response: Information has been provided accordingly (Lines 122-125).

Comment 5: 2.5. Dynamic rheological analysis: add citation

Response: Citation was added accordingly (Line 137).

Comment 6: 2.7. Sodium dodecyl sulfate-polyacrylamide gel electrophoresis (SDS-PAGE): add ref

Response: Reference has been added (Line 160).

Comment 7: Very weak experimental analysis

No experiments on gel forming ability; GEL STABILITY ETC

No experiment on foaming capacity

This study is suitable for short communication.

Results and Discussion

Since the methodology is very weak, results are not very interesting. Authors used a few experiments and performed correlation analysis. At least authors should perform the PCA and HCA analysis to discriminate the samples with or without addition of enzymes.

Food application of gel is also imp. Authors performed no studies in this section.

Overall, this study is incomplete with fewer experiments.

Response: The main purpose of the current study is to solve the texture problem of the gels made from protein isolate extracted from catfish byproducts. As mentioned in the Introduction, there has been no research on the application of MTGase to highly denatured fish protein isolate. Therefore, this study focused mainly on the technical feasibility of the enzymatic technology.

We are grateful for the valuable comments from the reviewers and will adopt them in our following research on protein isolate, including its functionalities for various food applications.  

Round 2

Reviewer 2 Report

I have no further comment.

Author Response

Thank you!

Reviewer 3 Report

The authors have given some explanations. However, no explanation was given for statistical analysis. This part needs to be elaborated. In addition, it will become more understandable if  sampling subtitle is opened and which analyzes are made in which samples.

Author Response

More explanation was added for the statictical analysis (Lines 176-181).

Reviewer 4 Report

Authors failed to add suggestions made by me. In my opinion, this version still many experimental flaws. Most of my suggestions are ignore without appropriate response.

Very weak experimental analysis

No experiments on gel forming ability; GEL STABILITY ETC

No experiment on foaming capacity

This study is suitable for short communication.

Authors used a few experiments and performed correlation analysis. At least authors should perform the PCA and HCA analysis to discriminate the samples with or without addition of enzymes.

I also raised these questions during first review; however, no response from authors. This version has less novelty.

Author Response

We appreciate the reviewer's comments.